# An In Vitro Study of the Effect of CO_2_ Laser Power Output on Ablative Properties in Porcine Tongue

**DOI:** 10.3390/life13010162

**Published:** 2023-01-05

**Authors:** Amontep Mungmee, Sajee Sattayut

**Affiliations:** 1Department of Oral and Maxillofacial Surgery, Khon Kaen University, Khon Kaen 40002, Thailand; 2Lasers in Dentistry Research Group, Khon Kaen University, Khon Kaen 40002, Thailand

**Keywords:** power output, carbon dioxide laser, ablative properties, oral soft tissue

## Abstract

CO_2_ lasers have been generally used in oral soft-tissue surgery. Although an increase in power enhances the depth of ablation, the specific correlation between units of power and ablative depth has not been stated. This study aimed to explore ablative depth and width affected by a power unit of 10,600-nm CO_2_ laser continuous wave at 3 to 10 W in tissue blocks of the swine tongue. The 112 samples were randomly allocated into eight groups according to the power settings. The depth and width of the incision were measured by using the ImageJ program. The 3 W group showed the minimal depth of ablation [0.527 mm (0.474 and 0.817)] and width of ablation [0.147 mm (0.110 to 0.184)]. The maximal depth [3.750 mm (3.362 and 4.118)] and width [0.700 mm (0.541 to 0.860)] were in the 9 W group. The correlation between the ablative depth and power unit was 0.81 (*p* < 0.001). Based on the regression equation (*p* < 0.001), the ablative depth (mm) = (the power unit of laser in W × 0.491) − 0.731. In conclusion, the ablative depth of the CO_2_ laser on soft tissue was strongly correlated to power units enabling the prediction equation.

## 1. Introduction

CO_2_ lasers have been reputed for their utility in intraoral soft tissue surgery, in particular for the removal of oral mucosal lesions [1]. The recent review of related articles of literature found that applications of CO_2_ lasers in oral surgery included operations on potentially malignant lesions, such as oral lichen planus and leukoplakia [2,3,4,5,6,7,8,9], benign tumors, such as oral papilloma, epulis, traumatic fibroma [2,10,11,12] and even malignant lesions, such as carcinoma in situ [2]. The outcomes of CO_2_ laser surgery are satisfactory as follows: time-saving procedure, reduced blood loss due to its hemostasis properties, less pain and discomfort, and no significant complications or recurrences compared with other modalities. [1,8,9,11,12,13] A recent study by Lazăr et al. [14] revealed the laser-enhanced proliferation rate of osteoblasts and their cellular activity. Another interesting aspect was that the laser also reduced oxidative stress in periodontal disease and that the use of a laser with periodontal therapy, such as scaling and root planning, improved the clinical outcome [15].

Regarding laser surgery, photoablation occurs by tissue absorption of the laser. The 10,600-nm CO_2_ laser is well absorbed by water. Therefore, oral soft tissue, which is composed predominantly of water, absorbs energy from the laser, resulting in an increase in temperature. When the water in the tissue evaporates at 100 °C, it causes soft tissue ablation, producing a vaporized zone. This photothermal reaction or photothermolysis is the main photoreaction of CO_2_ laser surgery. The vaporized zone is surrounded by an irreversible necrotic tissue zone, denatured proteins, and a photoactivation area or reactionary zone. The denatured protein of the coagulative zone is reversible tissue damage. The outer zone of photoactivation is affected by photobiomodulation, which enhances wound healing. When the soft tissue temperature reaches more than approximately 200 °C, a carbonized zone will occur on the ablative surface [1]. Due to its properties, the CO_2_ laser is suitable for oral soft tissue surgery. Regarding laser irradiation to the carbonized area on the surface of the laser incision, the laser energy will be absorbed more into the carbon than the tissue surface [16,17]. This results in more difficulty in tissue ablation than the first attempt. Besides, less clinically favorable healing may be found due to the excessive thermal effect. Thus, it is important for the CO_2_ laser parameter setting to achieve sufficient depth of ablation within a single passage.

There was a study by Wilder et al. [18] that showed that the ablative depth of a 9300-nm CO_2_ laser correlated strongly with the average power. This was also confirmed by Jaleel et al. [19] using a 10,600-nm CO_2_ laser. Whilst an increase in power was known to enhance the depth of ablation, the specific relation between units of power and ablative depth has not been explored enough to predict the depth of ablation by using power units. This exploration will enable an opportunity to predict the ablative depth by using a laser power unit, which is beneficial for adjusting the power for the desired depth of the laser incising soft tissue. Besides the effectiveness of laser ablation, minimized thermal damage will be achieved via a single incision. In order to perform a single incision by using a laser, an appropriate power setting unit, in relation to the ablative depth, is a crucial factor in performing oral soft tissue surgery. Since the prediction of the ablative depth by the power unit setting has not been proposed, the exploration of this prediction is a novelty and will benefit clinical practice.

Regarding the measurement of the depth of the incision, there was a histopathological exploration of thermal change for the assessment of ablative depth and width [18,20]. This measurement—via microscopic finding—enables us to evaluate affected zones due to the thermal change of the specimen. Owing to the histological process, the depth and width of the vaporized zone of this technique may not represent the ablative zone of the fresh tissue. There was a method of immediately measuring the depth of laser incision in fresh soft tissue after irradiation by using an endodontic file with a size of 80. It claimed to analyze the ablative zone of fresh specimens [19]. By using software for imaging analysis, the ImageJ program—a freeware for research from the National Institute of Health—can provide a variety of measurements and analyses of photographs. In the published research, this analysis showed high repeatability and validity of measurement for the assessment of the size of ulcers and the angulation of leg joints in clinical trials [21,22]. Therefore, the photographic analysis measurement by using the ImageJ software is applied to assess the immediate effect of laser ablation on fresh specimens.

This in vitro experiment aimed to compare the ablative depth and width of a 10,600-nm CO_2_ laser with a power setting based on a therapeutic range of 3 to 10 W in a soft tissue swine tongue sample. The correlation and regression between power units and vaporized zones were also explored. The prediction equation of the ablative depth by power unit was stated.

## 2. Materials and Methods

The in vitro experiment was conducted on 112 samples of tissue blocks. There were eight groups based on laser power settings from 3 W to 10 W of a CO_2_ laser at continuous wave.

### 2.1. Sample Estimation and Allocation

The sample size estimation of each group for comparing the depth and width of ablation was calculated by using the data from the previous study by Wilder-Smith et al. [18]. The formulation for sample size estimation is as follows:n/gr = [2(Z_α/2_ + Z_β_)^2^*δ^2^]/Δ^2^

α = 0.05, β = 0.2, δ = 0.108: variance of the depth of incision from the Wilder-Smith et al. study and Δ = the difference of incisional depth between the groups = 0.115 mm.

Regarding the calculation, the sample size of this study was 14 samples in each group. There were eight groups of CO_2_ power parameters power settings from 3 W to 10 W of CO_2_ laser at continuous wave. Therefore, the total number of samples in this in vitro experiment was 112.

In order to allocate the samples equally into each group, block randomization was performed.

### 2.2. Sample Preparation

We followed the methods of other published articles in exploring the ablative effect of a laser on oral soft tissue [18,19,23,24,25]. The samples were prepared from fresh organs of animals which were stored in the freezer after they were removed, and an experiment was undertaken in 24 h. Regarding the selection of the type of tissue, we decided to use tissue blocks prepared from fresh swine tongues, as undertaken in other studies exploring the ablative properties of a high-intensity laser [23,24,26]. Moreover, the porcine oral mucosa not only appears clinically but also in histology and immunochemistry, similar to human oral mucosa [27]. In order to prevent cell autolysis, the tongues were frozen at 4 °C immediately after they were removed. All specimens were placed in a temperature-controlled environment until they reached room temperature at 25 °C. A standard tissue block was made in the size of 2 × 1 × 2 cm from the ventral side of the swine tongue. The ventral surface mucosa was used as an experimental area. The experiment was conducted within 24 h after the animals were sacrificed. Based on this preparation, it was not shown tissue necrosis or deformity of oral epithelium and submucosal connective tissue in histological sections of an in vitro experiment exploring the ablative effects of the CO_2_ laser [28].

### 2.3. Laser Parameters

The laser machine used in this in vitro study was the 10,600-nm CO_2_ laser; Carbon Dioxide Model, Sphere FX Phototherapy units, Model: ST-2500, Thailand. The delivery system was an articulated arm. The handpiece provided a spot size of 0.2 mm in diameter and a focal length of 20 mm. The frequency of emission was a continuous wave. The power settings were 3 W (Power Density = 9548.1 W/cm^2^), 4 W (Power Density =12,730.7 W/cm^2^), 5 W (Power Density = 15,913.4 W/cm^2^), 6 W (Power Density =19,096.1 W/cm^2^), 7 W (Power Density = 22,278.8W/cm^2^), 8 W (Power Density = 25,461.5W/cm^2^), 9 W (Power Density = 28,644.2 W/cm^2^), 10 W (Power Density = 31,826.9W/cm^2^).

### 2.4. Method

The tissue sample covered with the intact mucosa of the ventral side of the tongue was placed into the slot of the customized apparatus as shown in Figure 1 and Figure 2. The tension was applied on both sides of the sample by using 100-g pendulums connected with the soft tissue hooks as shown in Figure 3 and Figure 4. This mimicked tension on surface tissue during CO_2_ laser ablation in a clinical situation. The sample was then ablated randomly by the 3 to 10 W CO_2_ laser at the midline of the tissue surface parallelly to the tension hooks with controlled linear movement at 2.5 mm/s. The length of ablation was 1 cm. External evacuation (COXO C-AS first-generation, Guangdong Province, China) was used during ablation.

After irradiation, the tension hooks were released from the specimen. The sample was immediately photographed from the side at a fixed distance and photography settings. The camera was placed on a tripod with a fixed angle, fixed distance between the specimen and the lens, and fixed focal length for photographing. The depth and width of the incision were measured by using the ImageJ program.

### 2.5. Outcome Measurement

The actual depth and width of the CO_2_ laser ablating tissues were measured from the recorded photographs by processing via the ImageJ program. The photographs from the lateral view were randomly measured for depth and width of ablation, respectively. The depth of ablation was an average of the distance from the tissue surface to the bottom of the incision on both sides, as shown in Figure 5. The width of ablation was measured at the bottom of the incision, as shown in Figure 6. To calibrate the measurement method, intra-reliability observation was also conducted. The reliability of the measurement was checked by comparing the ablative depth and width measured in the same manner by the researcher and another observer.

### 2.6. Statistical Analyses

Data were analyzed by SPSS/PC Evaluation Version 27.0. (Armonk, NY, USA: IBM Corp).

The data were described using descriptive statistics, including the means of ablative depth and width and their 95% confidence intervals. The ablative depth and width among the groups were compared using ANOVA with post hoc comparison (Tukey HSD) between the groups. In this case, the data was not a normal distribution. Thus, the median and the Kruskal-Wallis test were used as descriptive statistics and analytical statistics, respectively. The significance level was set at α = 0.05. The intraclass correlation coefficient (ICC) was performed. The interpretation was as follows; poor (<0.5), fair (0.5–0.74), good (0.75–0.9), and excellent (>0.9).

The relation between laser power unit and ablation, ablative depth, and width were analyzed by using Pearson and Spearman correlation coefficient. Linear regression was also calculated between the laser power unit and ablative distance.

## 3. Results

The 112 samples were ablated by a CO_2_ laser. There was no destructed sample. The results were as follows:

### 3.1. Reliability Test

The ICC and 95% confidence interval for intra- examiner reliability for ablative width and depth were between good 0.864 (0.513 to 0.967) and excellent 0.992 (0.967 to 0.998).

### 3.2. Comparing Ablative Depth by Different Power Settings of a CO_2_ Laser

Determining the distribution of ablative depth from all 112 specimens, the Shapiro-Wilk test represented the evidence of normality (*p* = 0.067), while the ablative depth of the 4 W group was not normally distributed (*p* = 0.029). Therefore, the median was used to describe the data in Table 1 and Figure 7. The minimal depth of incision was found in the 3 W group (median = 0.527 mm), whereas the maximal depth was found in the 9 W group (median = 3.75 mm). The depth of ablation tended to increase by increasing the laser power unit apart from the 7 W group and the 10 W group.

The Kruskal Wallis test illustrated that the 3 W group showed significantly less depth of ablation than the group irradiated by 6 to 10 W (*p* < 0.05). There was no statistically significant difference in the depth of ablation among the groups irradiated by 3 to 5 W and among the groups irradiated by 6 to 10 W, as shown in Table 2. The results enabled us to categorize the laser power outputs into three groups. The lower power group was made up of groups 3 to 4 W, which was statistically different from the higher power group comprising the groups of 7 to 10 W.

### 3.3. Relation between the Power Unit of a CO_2_ Laser and Ablative Depth

The results of the Pearson correlation showed that there was a strongly significant positive correlation between the laser power unit and ablative depth (r = 0.81 and *p* < 0.001). The linear regression of the laser power unit prediction of ablative depth was y = (0.491) x-0.731 (*p* < 0.001), where x was the power unit in W and y was ablative depth in mm. Therefore, the equation for the prediction of the depth of CO_2_ laser ablation was as below.
Depth of 10,600 nm CO_2_ laser = (power unit of laser × 0.491) − 0.731

### 3.4. Comparing Ablative Width by Different Power Settings of a CO_2_ Laser

The data of ablative width from all samples showed non-normal distribution at *p* < 0.001. The ablative width of each group was in normal distribution at *p* > 0.05. Therefore, the mean was used to describe the data in Table 3 and Figure 8. Like ablative depth, the minimal ablative width was found in the 3 W group (0.147 mm ± 0.064), and the maximal depth was in the 9 W group (0.700 mm ± 0.276). The width of ablation also tended to increase by increasing the laser power unit apart from the 5 W group, 7 W group, and the 10 W group. By comparison, the 9 W group with the 3 to 7 W groups revealed statistically significant differences at *p* < 0.05. There were no statistically significant differences among the 3 to 5 W groups, the 6 to 8 W groups, and the 9 to 10 W groups, as shown in Table 4.

### 3.5. Relation between the Power Unit of a CO_2_ Laser and the Ablative Width

The correlation between the laser power unit and the ablative width was not able to be computed because the data from all of the samples were not in the normal distribution.

## 4. Discussion

In our study, the porcine tongue specimens were ablated by a 10,600 nm CO_2_ laser at a power of 3 to 10 W, which provided power density in the range of 9545.5 to 31,818.2 w/cm^2^. The ablative depth and width were between 0.527 and 3.750 mm, and 0.147 and 0.7 mm, respectively. It was noticed that the laser power output at 9 W showed the deepest and widest incision rather than the highest power group of 10 W. Our results were found to be similar to some groups in the study of Beer et al. [29] in that the incision depth made by the 980-nm diode laser at an average power of 2.5 W in micropulse mode at a speed of 1 mm/s was deeper than 3.5 W and 4.5 W. Moreover, the incision depth at an average power of 3.5 W in pulse mode at a speed of 0.5 mm/s was deeper than 4.5 W. Those results may be explained by a limited thermal reaction due to tissue absorption. These phenomena were also found in the optical absorptivity of metal which slightly decreased with increasing laser power [30]. Thereby, increasing power may not definitely raise the ablative depth. Although the ablative depth in 9 W was deeper than 10 W, there was no statistical difference between the two groups.

In comparison, Jaleel et al. [19] used a 10,600 nm CO_2_ laser continuous wave with a power density in the range of 9554.1 to 15,923.6 W/cm^2^, and their means of ablative depth (4.5 to 6 mm) and width (2 to 3.5 mm) were larger than in our study. Regarding the point of measurement, the width of ablation in the study by Jaleel et al. was measured at the surface of the incision, whilst in our study, this was measured at the bottom of the incision, representing an ablative width with less tension. This is contrary to Wilder-Smith et al.’s study [18], which used a 9300 nm CO_2_ laser continuous wave with a power of 3.5 to 9 W and a pulse width of 1 to 200 ms. The ablative depth (0.327 to 1.075 mm) and width (0.134 to 0.372 mm) in Wilder-Smith et al.’s study were less than in our study. This was explained by the higher absorption coefficient of 10,600 nm in water compared with the 9300 nm CO_2_ laser. However, the measurements of those studies were different. In the study of Jaleel et al., an endodontic file number 80 with a rubber stop was used to mark the distance of ablative depth and width. From our experiences, this measurement method was convenient, but it was difficult to control the endodontic file from unintentionally penetrating the peripheral soft tissue. This may also result in an over-record of ablative depth and width measurement. The measurement of outcome from the study of Wilder-Smith et al. was based on histological sections. This could provide an advantage in evaluating tissue alteration adjacent to the ablative zone. However, multiple processes of histological sectioning and staining would distort the specimens. The measurement from the histological section may not represent the actual incision of the fresh sample. Our study, therefore, was the first to introduce the measurement of actual ablative depth and width in oral soft tissue specimens using the ImageJ program. There were advantages of the application of the ImageJ program, which was a digital processor analyzer. Firstly, it was convenient for recording data in the image file, which was available for repeated measurements. Secondly, the record for measuring was immediately taken after the specimens were irradiated. This provided minimal distortion of the incisional zone of the sample. Our method of measurement had a high intra-reliability of over 80 percent, as shown in the results of our study. We also recommended the immediate measurement of the effect of the CO_2_ laser in the soft tissue specimens using a fixed photography setting and the ImageJ program.

The relationship between the power output unit and ablative distances was found in our study, similar to the other studies [18,19]. The higher the average laser power output, the more ablative depth and width increased. The results of our study were able to categorize the power unit settings by ablative depth into three groups: low power (3 to 4 W, ablative depth between 0.527 to 0.996 mm), medium power (5 to 6 W, ablative depth between 1.842 to 2.507 mm) and high power (7 to 10 W, ablative depth between 2.403 to 3.750 mm). This finding provided benefits for oral surgeons in practice. The surgeons should adjust the CO_2_ power unit corresponding to the desired depth of ablation and the groups of power units rather than only increasing by 1 W until achieving the depth of the incision.

The strong correlation between the laser power unit and ablative depth was also found in our study at 0.81 (*p* < 0.01). This correlation was higher than the study of Wilder-Smith, which was 0.46. Our study was the first to formulate the equation of ablative depth by the power setting unit. It should be emphasized that this prediction was based on a controlled linear speed of 0.25 cm/sec with surface tension at 100 g.

Regarding the surgical techniques of a CO_2_ laser for treating potentially malignant disorders, Nammour et al. [6] postulated the highest clinical outcome of applying a CO_2_ laser for complete excision by extending 1 mm in depth and 3 mm into the surrounding healthy tissue for removal. The prediction of ablative depth from our study enables oral and maxillofacial surgeons to set a power unit of 10,600 nm CO_2_ laser effectively.

Ablative width was also important; based on the study of Holmstrup et al. [31], surgical excision of oral premalignant lesions with a scalpel should include 3 to 5 mm of clinically normal mucosa surrounding the lesion. Romeo et al. [32] also reported the success of 10,600 nm CO_2_ laser excision of oral leukoplakia by using a power of 4.5 W with at least 3 mm extending from the margin. In terms of application from our study, using a CO_2_ laser at 4 to 5 W should be combined with an overextension of about 0.3 mm due to the range of ablative width (0.208 to 0.29 mm).

The controlled depth of ablation to avoid overextension is also important in some operations when applying a CO_2_ laser. Using a CO_2_ laser for vestibuloplasty [33,34] should be cautious of preserving the periosteum avoiding complications of bony exposure and delayed healing. When Kongsong et al., 2020 [35] treated trigeminal neuralgia with a CO_2_ laser, a supra periosteal flap was elevated, then an affected nerve was ablated with the laser. The depth of CO_2_ laser ablation must be limited to avoid injury of vital adjacent structures, such as the nerve, the vessel, and the periosteum. Therefore, the desired ablative depth is essential in clinical application, as mentioned.

As far as our study is concerned, it was conducted in fresh swine tongue tissue blocks, which lack vascularization or blood flow. From Kinikoglu et al. study [27], the composition of swine oral mucosa is similar to human mucosa, and most of the CO_2_ laser was absorbed by water, not blood, in the vessels. Then the lack of vascularization or blood flow is not a major factor in altering the results from the prediction equation of ablative depth. Moreover, based on our study, the maximal ablative depth in the 9 W group was approximately 3.7 mm. In human oral mucosa, the mean tissue thickness was 3.8 mm [36], and received blood perfusion mostly by capillary (low blood flow). As far as the vascularization in the tissue block is concerned, which is not necessarily vital tissue, the limitation of the prediction found in this study was still applied in non-inflamed mucosa. Unlike non-inflamed mucosa, the inflammatory mucosa composed of numerous vessels and redundant cellular fluid is not the same composition of water and hemoglobin. In highly vascularized tissue, then, the prediction equation of ablative depth should be used with caution.

## 5. Conclusions

Based on our results, the depth and width of using 10,600 nm CO_2_ laser ablation in the range of 3 to 9 W were increased by increasing power units. According to the power output settings, these were categorized into three groups, namely, the lower power group at 3 to 4 W, the middle power group at 5 to 6 W, and the higher power group at 7 to 10 W. The ablative depth of the CO_2_ laser was strongly positively correlated with the laser power output and was predicted by power units.

## Figures and Tables

**Figure 1 life-13-00162-f001:**
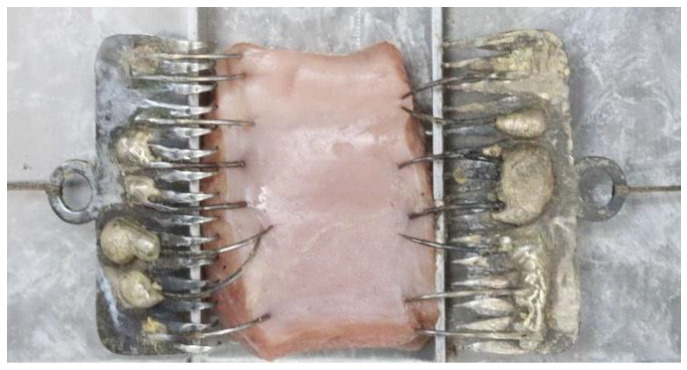
The specimen was placed into the slot (top view).

**Figure 2 life-13-00162-f002:**
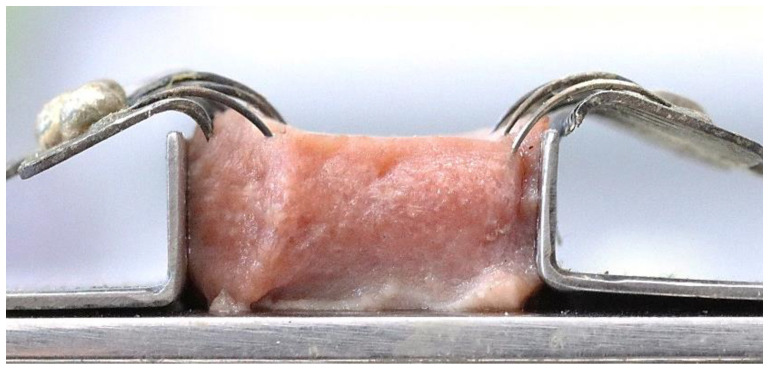
The specimen from a side view.

**Figure 3 life-13-00162-f003:**
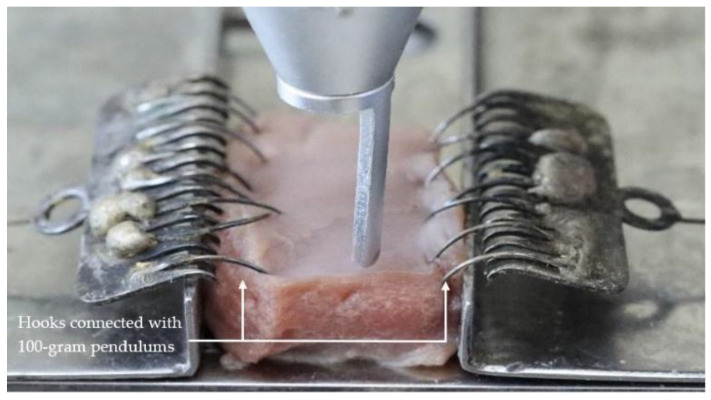
The specimen and laser probe were attached to the customized apparatus (top view).

**Figure 4 life-13-00162-f004:**
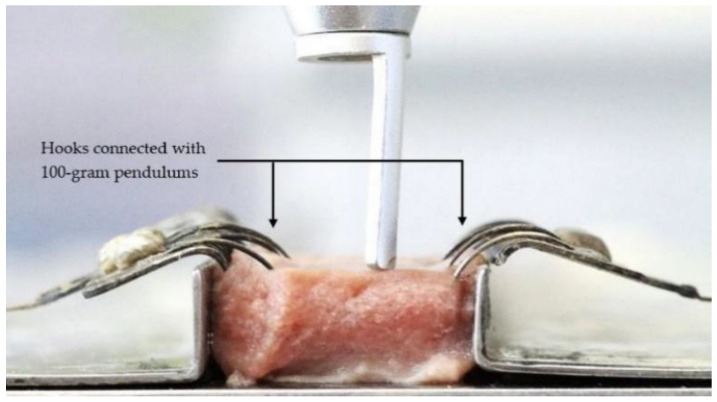
The Specimen and laser probe were attached to customized apparatus (side view).

**Figure 5 life-13-00162-f005:**
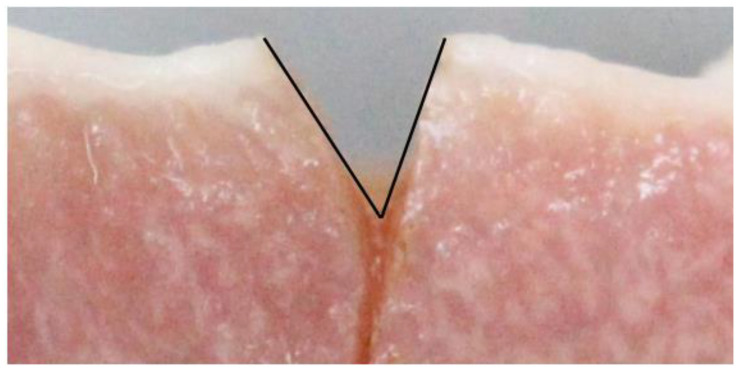
The lines for measuring the depth of ablation by using the ImageJ program.

**Figure 6 life-13-00162-f006:**
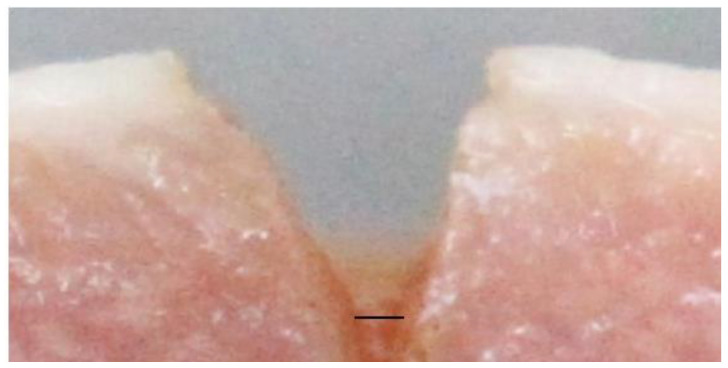
The line for measuring the width of ablation by using the ImageJ program.

**Figure 7 life-13-00162-f007:**
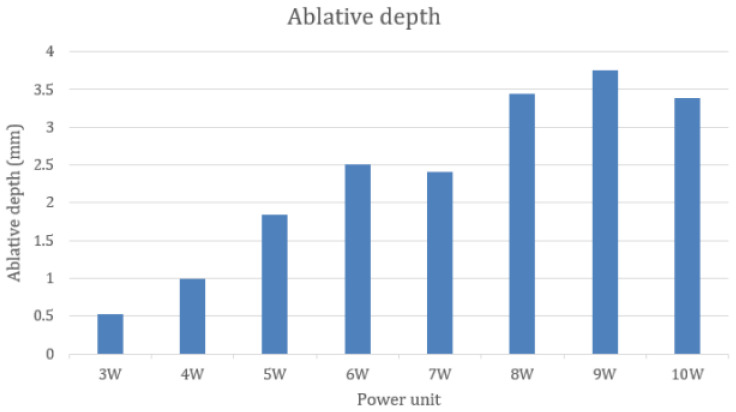
The median of ablative depth by groups.

**Figure 8 life-13-00162-f008:**
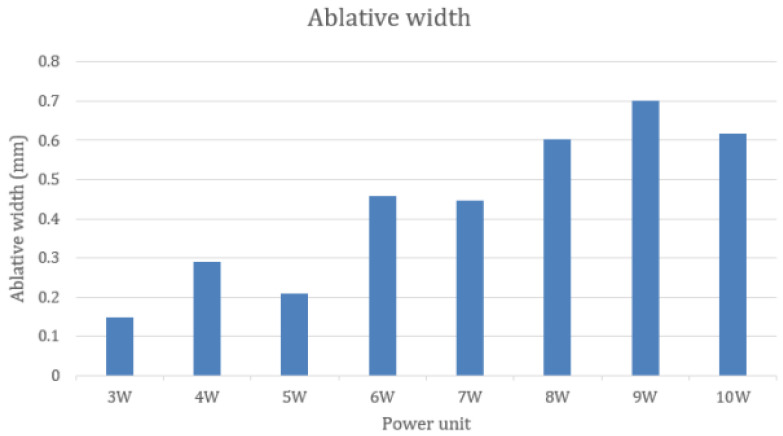
The ablative width by the groups.

**Table 1 life-13-00162-t001:** The ablative depth by the groups.

The Depth of the Incision								
	3 W	4 W	5 W	6 W	7 W	8 W	9 W	10 W
**Median (mm)**	0.527	0.996	1.842	2.507	2.403	3.447	3.750	3.388
**25th percen tile**	0.474	0.842	1.326	2.063	3.362	2.589	3.362	3.151
**75th percen tile**	0.817	1.205	2.199	2.705	4.118	4.870	4.118	4.453

**Table 2 life-13-00162-t002:** Comparisons of the ablative depth between the groups.

The Pairwise Comparisons of the Ablative Depth							
	4 W	5 W	6 W	7 W	8 W	9 W	10 W
**3 W**	*1*	*0.283*	*0.002*	*<0.001*	*<0.001*	*<0.001*	*<0.001*
**4 W**		*1*	*0.098*	*0.021*	*<0.001*	*<0.001*	*<0.001*
**5 W**			*1*	*1*	*0.017*	*0.002*	*0.009*
**6 W**				*1*	*1*	*0.269*	*0.89*
**7 W**					*1*	*0.916*	*1*
**8 W**						*1*	*1*
**9 W**							*1*

Data are presented in *p*-value.

**Table 3 life-13-00162-t003:** The ablative width by the groups.

The Mean and 95% Confidence Interval	
**3 W**	0.147 (0.110–0.184)
**4 W**	0.290 (0.198–0.382)
**5 W**	0.208 (0.163–0.254)
**6 W**	0.458 (0.366–0.550)
**7 W**	0.446 (0.322–0.570)
**8 W**	0.602 (0.469–0.735)
**9 W**	0.700 (0.541–0.860)
**10 W**	0.618 (0.452–0.785)

Data present in millimeters.

**Table 4 life-13-00162-t004:** Comparisons of the ablative width between the groups.

The Tukey Post-Hoc Test of Ablative Width							
	4 W	5 W	6 W	7 W	8 W	9 W	10 W
**3 W**	*0.559*	*0.992*	*0.002*	*0.003*	*<0.001*	*<0.001*	*<0.001*
**4 W**		*0.960*	*0.345*	*0.444*	*0.002*	*<0.001*	*0.001*
**5 W**			*0.028*	*0.044*	*<0.001*	*<0.001*	*<0.001*
**6 W**				*1*	*0.548*	*0.037*	*0.408*
**7 W**					*0.441*	*0.023*	*0.313*
**8 W**						*0.898*	*1*
**9 W**							*0.959*

Data are presented in *p*-value.

## Data Availability

The data presented in this study are available on request from the corresponding author.

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
