# Peer review of "An In Vitro Study of the Effect of CO2 Laser Power Output on Ablative Properties in Porcine Tongue"

_life, 2023, doi:10.3390/life13010162_

Round 1
Reviewer 1 Report
Major.
It is already well established that the depth of the ablation is proportional to the output power or more accurately to the energy density delivered. Unfortunately, there is no additional input obtained from this manuscript. Therefore, it cannot be accepted to be published in Life Journal, MDPI.
Author Response
Response to the reviewers
Reviewer 1
Point 1
It is already well established that the depth of the ablation is proportional to the output power or more accurately to the energy density delivered. Unfortunately, there is no additional input obtained from this manuscript. Therefore, it cannot be accepted to be published in Life Journal, MDPI.
Response to point 1
Regarding the novelty of our work, we wrote some sentences to emphasize about the novelty of this research. We already stated the research gap in the line 54-64 “Whilst an increase in power was known to enhance the depth of ablation, the specific relation between units of power and ablative depth has not been enough exploration to predict the depth of ablation by using power units. This exploration will enable an opportunity to predict the ablative depth by using a laser power unit, which is beneficial to adjust the power for the desired depth of the laser incising soft tissue. Besides the effectiveness of laser ablation, minimized thermal damage will be achieved via a single incision. In order to perform a single incision by using laser, an appropriate power setting unit in relation to the ablative depth is a crucial factor in performing oral soft tissue surgery. Since the prediction of the ablative depth by the power unit setting has not been purposed. The exploration of this prediction is a novelty and benefit in clinical practice.”
We stated more important objective about prediction of ablative depth with power unit as follows:- The prediction equation of the ablative depth by power unit was stated. We also stated more details in the discussion about the novelty of this research finding in line 300-302 “Our study, therefore, was the first to introduce the measurement of actual ablative depth and width in oral soft tissue specimens using the ImageJ program”
We hopefully you agree the novelty in our research.
"Please see the attachment"

Reviewer 2 Report
Interesting topic and well written paper. However, in vitro results, although the pig tongue is fresh, are not similar with in vivo ones (lack of vascularizations, ecc). Limitations of the study should be considered. Authors have to amplify the novelty of the study, which I can't find.
Author Response
Response to the reviewers
Reviewer 2
Point 1 However, in vitro results, although the pig tongue is fresh, are not similar with in vivo ones (lack of vascularizations, ecc). Limitations of the study should be considered.
Response to point 1
Thank you for your advice, we added the limitation of this study in last paragraph of the discussion in line 344-357. “As far as the concern of our study conducting in fresh swine tongue tissue blocks, which lack of vascularization or blood flow. From Kinikoglu et al study [27], the composition of swine oral mucosa is similar to human mucosa, and most of CO2 laser was absorbed by water not blood in the vessels. Then the lack of vascularization or blood flow is not a major factor to alter the results from the predication equation of ablative depth. Moreover, based on our study, the maximal ablative depth in 9 W group was approximately 3.7 mm. In human oral mucosa, the mean tissue thickness was 3.8 mm [36], received blood perfusion mostly by capillary (low blood flow). As far as the vascularization concerned in the tissue block which is not reassembly the vital tissue, the limitation of the prediction found in this study still was applied in non-inflamed mucosa. Unlike, the inflammatory mucosa composed of numerous vessels and redundant cellular fluid is not the same composition of water and hemoglobin by comparison with normal mucosa. Then in high vascularized tissue, the prediction equation of ablative depth should be used with caution.”
Point 2 Authors have to amplify the novelty of the study, which I can't find.
Response to point 2
Regarding the novelty of our work, we wrote some sentences to emphasize about the novelty of this research. We already stated the research gap in the line 54-64 “Whilst an increase in power was known to enhance the depth of ablation, the specific relation between units of power and ablative depth has not been enough exploration to predict the depth of ablation by using power units. This exploration will enable an opportunity to predict the ablative depth by using a laser power unit, which is beneficial to adjust the power for the desired depth of the laser incising soft tissue. Besides the effectiveness of laser ablation, minimized thermal damage will be achieved via a single incision. In order to perform a single incision by using laser, an appropriate power setting unit in relation to the ablative depth is a crucial factor in performing oral soft tissue surgery. Since the prediction of the ablative depth by the power unit setting has not been purposed. The exploration of this prediction is a novelty and benefit in clinical practice.”
We stated more important objective about prediction of ablative depth with power unit as follows:- The prediction equation of the ablative depth by power unit was stated. We also stated more details in the discussion about the novelty of this research finding in line 300-302 “Our study, therefore, was the first to introduce the measurement of actual ablative depth and width in oral soft tissue specimens using the ImageJ program”
We hopefully you agree the novelty in our research.
"Please see the attachment"

Reviewer 3 Report
The conclusion of the abstract is not very well illustrating the results
introduction
line 26 I propose the change of precancerous lesions with potentially malignant lesions
line 48 reference after the name of the author Wilder et al
Materials and Methods
explain why the size of the group is 14 specimens
describe the calibration of the measure for image J program
Discussion
illustration of the result of predilection of depth and width of the 10 watts when 9 W have the highest
why a comparison of the three group not done
Author Response
Response to the reviewers
Reviewer 3
Point 1
line 26 I propose the change of precancerous lesions with potentially malignant lesions
line 48 reference after the name of the author Wilder et al
Response to point 1
Thank you for your suggestion. We corrected this word and reference as your suggestion.
Point 2
explain why the size of the group is 14 specimens
Response to point 2
We accepted your suggestion to describe more about sample size estimation which in section 2.1 . The formula and calculation of sample size were shown in manuscript.
Point 3
describe the calibration of the measure for image J program
Response to point 3
To calibrate the measurement, the intra-reliability observation was conducted. Researcher and another observer measured the depth and width in same manner and compare the result. (line 172-175) The intra-reliability observation for ablative width and depth were between good 0.864 (0.513 to 0.967) and excellent 0.992 (0.967 to 0.998) as show in result.
Point 4
illustration of the result of predilection of depth and width of the 10 watts when 9 W have the highest
Response to point 4
Thank you for your suggestion, we discuss more about power setting in 9 and 10 W groups and the ablative depth in line 270 to 280. “It was noticed that the laser power output at 9 W showed the deepest and widest incision rather than the highest power group of 10 W. Our results were found similarly to some groups in the study of Beer et al [29] in that the incision depth made by the diode laser at average power of 2.5 W in micropulsed mode in speed of 1 mm/s was deeper than 3.5 W and 4.5 W. Moreover, the incision depth at average power 3.5 W in pulsed mode at speed 0.5 mm/s was deeper than 4.5 W. Those results may be explained by a limited thermal reaction due to tissue absorption. This phenomena was also found in the optical absorptivity of metal which slightly decreased as increasing laser power.[30] Thereby, increasing power may not definitely rise the ablative depth. Although the ablative depth in 9 W was deeper than 10 W, there was no statistically difference between 2 groups.”
Point 5
why a comparison of the three group not done
Response to point 5
Our objective in this study is comparison the ablative depth among laser power unit 3 to 10 W. Thus, first we compared among this groups. Then we found the statistically difference between 3 groups. By this reason we categorized in 3 groups. To compare between 3 groups is not our purpose in this study.
Point 6
The conclusion of the abstract is not very well illustrating the results
Response to point 6
We changed conclusion in abstract. Thank you for your suggestion.

Reviewer 4 Report
This in vitro study brings little new in terms of information.
The authors should stress the originality of the study, which in my opinion is lacking.
Please see attached pdf for a point-by-point analysis

Author Response
Response to the reviewers
Reviewer 4
Point 1
The conclusion should be reformulated so it is more focused on the overall results and bottom line of the study.
Response to point 1
Thank you for your suggestion, we reformulated conclusion in abstract.
Point 2
The authors should explain why they chose CO2 laser vs other types of lasers used in the oral cavity for this study.
Response to point 2
Due to CO2 laser property, which has well absorption in water and oral soft tissue compose mainly water. Then CO2 laser is widely used for oral soft tissue surgery. We explained in line 45-46
Point 3
The authors should also discuss other applications of laser in the oral cavity. I suggest
Response to point 3
Thank you for your kindly suggestion. We added these studies in line 31-34.
Point 4
Since this is an in vitro study the authors should stress the potential clinical importance of the study.
Response to point 4
Thank you for your advice. As we knew our study is in vitro study, then we clarified this point in discussion (limitation) and stated how our in vitro study can apply in clinical practice. (line 344-357)
Point 5
The authors should state the limitations of the study.
Response to point 5
Thank you for your advice, we added the limitation of this study in last paragraph of the discussion in line 344-357. “As far as the concern of our study conducting in fresh swine tongue tissue blocks, which lack of vascularization or blood flow. From Kinikoglu et al study [27], the composition of swine oral mucosa is similar to human mucosa, and most of CO2 laser was absorbed by water not blood in the vessels. Then the lack of vascularization or blood flow is not a major factor to alter the results from the predication equation of ablative depth. Moreover, based on our study, the maximal ablative depth in 9 W group was approximately 3.7 mm. In human oral mucosa, the mean tissue thickness was 3.8 mm [36], received blood perfusion mostly by capillary (low blood flow). As far as the vascularization concerned in the tissue block which is not reassembly the vital tissue, the limitation of the prediction found in this study still was applied in non-inflamed mucosa. Unlike, the inflammatory mucosa composed of numerous vessels and redundant cellular fluid is not the same composition of water and hemoglobin by comparison with normal mucosa. Then in high vascularized tissue, the prediction equation of ablative depth should be used with caution.”
"Please see the attachment"

Reviewer 5 Report
The paper is an ex-vivo study that aims to observe the ablative depth and width resulting from a 10600 nm “CO2 laser” in continuous mode at a range of powers from 3 to 10 W in tissue blocks of the swine tongue. The study is interesting and well-designed, and the manuscript is well-written. There are some issues that need to be considered. - In the abstract, the authors reported the minimal and maximal depth and width of ablation without their unit of measurement. It is recommended to put the units. In addition, it seems that there is an error in the reported minimal width of ablation on lines 14 and 15. A revision is needed of the numbers. - Since the introduction is the rational part of the manuscript, it would be better to include a more clear statement in the introduction section that demonstrates the research gap in the literature that pushed the researchers to perform this experiment in order to emphasize the novelty of the present study.
In the materials and methods, the authors didn’t declare clearly how the 112 samples were distributed in the 8 groups, and if they were distributed equally how the sample size was chosen (with power analysis?). Please mention clearly the distribution of samples in each group and how the sample size was chosen. - If there were some destructed samples during the experiment, it would be recommended to declare that, with mentioning the cause and in which group that was occurred, this is in order to share the experience with other researchers.
- The authors stated that on lines 85 and 86 “ This preparation proved in the in vitro experiments [23] that the oral epithelium and submucosal connective tissue were within normal limit.” It is recommended to describe that in a more clear way because it is not enough to mention the reference.
In the “2.3 Method” section, if it is possible, the authors can provide photos or designs that demonstrates the methods described in the section. - On table 1, there is a missing letter “T” at the second title of the table; “he median and 25, 75 percentiles of the depth of the incision". - The authors put in the manuscript only tables of the results without any graphs. It would be better to add graphs that demonstrate well the results. - In the discussion section, the authors stated that “It was noticed that the laser power output at 9 W showed the deepest and widest incision rather than the highest power group of 10 W”. This observation is very interesting that needs to be discussed more in detail, where the authors only commented on that in one statement (lines 195 - 197). - The authors stated that “Our study, therefore, was the first to introduce the measurement of actual ablative depth and width using the ImageJ program.” Are the authors sure about that? If yes, it is recommended to add a reference that proves that. If not, the level of confirmation should be decreased.
- At the end of the discussion section, the authors should add a paragraph about the limitations of the study that they have observed from this experience. This is to share with others their experience and to help the reader to interpret the results precisely.
Author Response
Response to the reviewers
Reviewer 5
Point 1
The paper is an ex-vivo study that aims to observe the ablative depth and width resulting from a 10600 nm “CO2 laser” in continuous mode at a range of powers from 3 to 10 W in tissue blocks of the swine tongue. The study is interesting and well-designed, and the manuscript is well-written. There are some issues that need to be considered. - In the abstract, the authors reported the minimal and maximal depth and width of ablation without their unit of measurement. It is recommended to put the units. In addition, it seems that there is an error in the reported minimal width of ablation on lines 14 and 15. A revision is needed of the numbers. - Since the introduction is the rational part of the manuscript, it would be better to include a more clear statement in the introduction section that demonstrates the research gap in the literature that pushed the researchers to perform this experiment in order to emphasize the novelty of the present study.
Response to point 1
Your kind suggestion is followed. Unit of measurement in mm was inserted into the abstract and minimal depth of ablation was also corrected. There correction of line 13- 15 from “The 3 W group showed the minimal depth of ablation [0.527 (0.474 and 0.817)] and width of ablation [0.147 (10 to 0.184)]. The maximal depth [3.750 (3.362 and 4.118)] and width [0.700 (0.541 to 0.860)] were in the 9 W group” is changed to “The 3 W group showed the minimal depth of ablation [0.527 mm (0.474 and 0.817)] and width of ablation [0.147 mm (0.110 to 0.184)]. The maximal depth [3.750 mm (3.362 and 4.118)] and width [0.700 mm (0.541 to 0.860)] were in the 9 W group.”
->Thank you for your suggestion to emphasize our novelty. We already stated the research gap in the line 54-64 “Whilst an increase in power was known to enhance the depth of ablation, the specific relation between units of power and ablative depth has not been enough exploration to predict the depth of ablation by using power units. This exploration will enable an opportunity to predict the ablative depth by using a laser power unit, which is beneficial to adjust the power for the desired depth of the laser incising soft tissue. Besides the effectiveness of laser ablation, minimized thermal damage will be achieved via a single incision. In order to perform a single incision by using laser, an appropriate power setting unit in relation to the ablative depth is a crucial factor in performing oral soft tissue surgery. Since the prediction of the ablative depth by the power unit setting has not been purposed. The exploration of this prediction is a novelty and benefit in clinical practice.”.
We stated more important objective about prediction of ablative depth with power unit as follows:- The prediction equation of the ablative depth by power unit was stated.
Point 2
In the materials and methods, the authors didn’t declare clearly how the 112 samples were distributed in the 8 groups, and if they were distributed equally how the sample size was chosen (with power analysis?). Please mention clearly the distribution of samples in each group and how the sample size was chosen. - If there were some destructed samples during the experiment, it would be recommended to declare that, with mentioning the cause and in which group that was occurred, this is in order to share the experience with other researchers.
Response to point 2
We accepted your suggestion to describe more about sample size estimation and allocation which in section 2.1 . The formula and calculation of sample size were shown in manuscript. All samples were allocated equally into each group, block randomization was performed.
Point 3
If there were some destructed samples during the experiment, it would be recommended to declare that, with mentioning the cause and in which group that was occurred, this is in order to share the experience with other researchers.
There was no destructed sample and this already added in the result.
Point 4
- The authors stated that on lines 85 and 86 “ This preparation proved in the in vitro experiments [23] that the oral epithelium and submucosal connective tissue were within normal limit.” It is recommended to describe that in a more clear way because it is not enough to mention the reference.
Response to point 4
From previous study, the study revealed that the histological finding no tissue necrosis or deformity of oral epithelium and submucosal connective tissue was observed. So, I adopted preparation method to our study. The modification of referencing is “Based on this preparation, it was not shown tissue necrosis or deformity of oral epithelium and submucosal connective tissue in histological sections of an in vitro experiment exploring ablative of CO2 laser.[28]”.
We also apply the same method same as other references.” The samples were prepared from fresh organ of animal which stored in the freeze after scarified and undertaken experiment in 24 hours. [18,19,23-25]”
And we added more reference that showed porcine mucosa is similar to human mucosa “the porcine oral mucosa is not only clinical appearance but also histology and immunochemistry similar to human oral mucosa. [27]”
Point 5
In the “2.3 Method” section, if it is possible, the authors can provide photos or designs that demonstrates the methods described in the section.
Response to point 5
The new pictures of specimen in the customized machine were provided in the method.
Point 6
- On table 1, there is a missing letter “T” at the second title of the table; “he median and 25, 75 percentiles of the depth of the incision".
Thank you for your correction.
- The authors put in the manuscript only tables of the results without any graphs. It would be better to add graphs that demonstrate well the results.
Thank you for your suggestion. I added graphs of ablative depth and width.
Point 7
- In the discussion section, the authors stated that “It was noticed that the laser power output at 9 W showed the deepest and widest incision rather than the highest power group of 10 W”. This observation is very interesting that needs to be discussed more in detail, where the authors only commented on that in one statement (lines 195 - 197).
Response to point 7
Thank you for your suggestion, we discuss more about power setting in 9 and 10 W groups and the ablative depth in line 270 to 280 “It was noticed that the laser power output at 9 W showed the deepest and widest incision rather than the highest power group of 10 W. Our results were found similarly to some groups in the study of Beer et al [29] in that the incision depth made by the diode laser at average power of 2.5 W in micropulsed mode in speed of 1 mm/s was deeper than 3.5 W and 4.5 W. Moreover, the incision depth at average power 3.5 W in pulsed mode at speed 0.5 mm/s was deeper than 4.5 W. Those results may be explained by a limited thermal reaction due to tissue absorption. This phenomena was also found in the optical absorptivity of metal which slightly decreased as increasing laser power.[30] Thereby, increasing power may not definitely rise the ablative depth. Although the ablative depth in 9 W was deeper than 10 W, there was no statistically difference between 2 groups.”
Point 8
- The authors stated that “Our study, therefore, was the first to introduce the measurement of actual ablative depth and width using the ImageJ program.” Are the authors sure about that? If yes, it is recommended to add a reference that proves that. If not, the level of confirmation should be decreased.
Response to point 8
Yes, from review literature and previous studies, none of these were assessed ablative depth by ImageJ program. Recently (20/12/2022), I searched in Pubmed with keyword “ImageJ and ablative”, result was no study which correlate ImageJ and ablative depth in soft tissue specimen.
Point 9
- At the end of the discussion section, the authors should add a paragraph about the limitations of the study that they have observed from this experience. This is to share with others their experience and to help the reader to interpret the results precisely.
Response to point 9
Thank you for your advice, we added the limitation of this study in last paragraph of the discussion in line 344-357. “As far as the concern of our study conducting in fresh swine tongue tissue blocks, which lack of vascularization or blood flow. From Kinikoglu et al study [27], the composition of swine oral mucosa is similar to human mucosa, and most of CO2 laser was absorbed by water not blood in the vessels. Then the lack of vascularization or blood flow is not a major factor to alter the results from the predication equation of ablative depth. Moreover, based on our study, the maximal ablative depth in 9 W group was approximately 3.7 mm. In human oral mucosa, the mean tissue thickness was 3.8 mm [36], received blood perfusion mostly by capillary (low blood flow). As far as the vascularization concerned in the tissue block which is not reassembly the vital tissue, the limitation of the prediction found in this study still was applied in non-inflamed mucosa. Unlike, the inflammatory mucosa composed of numerous vessels and redundant cellular fluid is not the same composition of water and hemoglobin by comparison with normal mucosa. Then in high vascularized tissue, the prediction equation of ablative depth should be used with caution.”

Round 2
Reviewer 1 Report
Dear authors. The necessary corrections were made. This manuscript can be accepted for publications.
Reviewer 2 Report
Thank you for your corrections, now the paper is suitable for the publication.
Reviewer 4 Report
The manuscript has been improved
Reviewer 5 Report
No suggestions. The paper can be accepted in this form